# MᴜʟᴛɪGPʀᴏᴍᴘᴛ for Multi-Task Pre-Training and Prompting on Graphs

## ABSTRACT

Graphs can inherently model interconnected objects on the Web, thereby facilitating a series of Web applications, such as web analyzing and content recommendation. Recently, Graph Neural Networks (GNNs) have emerged as a mainstream technique for graph representation learning. However, their efficacy within an end-to-end supervised framework is significantly tied to the availability of task-specific labels. To mitigate labeling costs and enhance robustness in few-shot settings, pre-training on self-supervised tasks has emerged as a promising method, while prompting has been proposed to further narrow the objective gap between pretext and downstream tasks. Although there has been some initial exploration of prompt-based learning on graphs, they primarily leverage a single pretext task, resulting in a limited subset of general knowledge that could be learned from the pre-training data. Hence, in this paper, we propose MᴜʟᴛɪGPʀᴏᴍᴘᴛ, a novel multi-task pre-training and prompting framework to exploit multiple pretext tasks for more comprehensive pre-trained knowledge. First, in pre-training, we design a set of *pretext tokens* to synergize multiple pretext tasks. Second, we propose a dual-prompt mechanism consisting of *composed* and *open* prompts to leverage task-specific and global pre-training knowledge, to guide downstream tasks in few-shot settings. Finally, we conduct extensive experiments on six public datasets to evaluate and analyze MᴜʟᴛɪGPʀᴏᴍᴘᴛ[1].

## CCS CONCEPTS

• **Information systems** → **Web mining**; **Data mining**; • **Computing methodologies** → **Learning latent representations**.

## KEYWORDS

Graph neural networks, prompting, pre-training, multi-task, few-shot learning.

**ACM Reference Format:**
Anonymous Author(s). 2018. MᴜʟᴛɪGPʀᴏᴍᴘᴛ for Multi-Task Pre-Training and Prompting on Graphs. In *Proceedings of Make sure to enter the correct conference title from your rights confirmation emai (Conference acronym 'XX).* ACM, New York, NY, USA, 12 pages. https://doi.org/XXXXXXX.XXXXXXX

---

[1]See https://anonymous.4open.science/r/MultiGPrompt for code & data for review.

---

*Conference acronym 'XX, June 03–05, 2018, Woodstock, NY*
© 2018 Association for Computing Machinery.
ACM ISBN 978-1-4503-XXXX-X/18/06...$15.00
https://doi.org/XXXXXXX.XXXXXXX

## 1 INTRODUCTION

The World Wide Web has evolved into an universal data repository, linking an expansive array of entities to create vast and intricate graphs. Mining such widespread graph data has fueled a myriad of Web applications, ranging from Web mining [1, 52] and social network analysis [59, 63] to content recommendation [34, 64]. Contemporary techniques for graph analysis predominantly rely on graph representation learning, particularly graph neural networks (GNNs) [15, 24, 43, 51]. Most GNNs operate on a message-passing framework, where each node updates its representation by iteratively receiving and aggregating messages from its neighbors [50], while more recent approaches have also explored transformer-based architectures [19, 54, 57].

**Pre-training.** GNNs are conventionally trained in an end-to-end manner, which heavily depends on the availability of large-scale, task-specific labeled data. To reduce the dependency on labeled data, there has been a growing emphasis on pre-training GNNs [17, 18, 30, 33], following the success of pre-training in natural language and computer vision domains [3, 7, 9]. Pre-training GNNs involves optimizing self-supervised pretext tasks on graphs without task-specific labels. Such pretext tasks are designed to capture intrinsic graph properties, such as node and edge features [17, 18, 55], node connectivity [15, 18, 30, 55], and local or global graph patterns [17, 30, 33, 43, 55]. Hence, pre-training yields a task-agnostic foundation that can be subsequently fine-tuned to a specific downstream task with a limited amount of labeled data. To further mitigate the inconsistency between pre-training and fine-tuning objectives [26], prompt-based learning has been first proposed in language models [5]. A prompt acts as an intermediary that reformulates the downstream task to align with pre-training, without the need to update the parameters of the pre-trained model. As a prompt entails much fewer parameters than the pre-trained model, it is especially amenable to few-shot settings where there are very limited task-specific labels.

Following the success of prompt-based learning, researchers have also begun to explore prompt-based or related parameter-efficient learning on graph data [11, 25, 29, 40–42]. However, most existing research in prompt-based graph learning only utilizes a single pretext task in pre-training. Not surprisingly, different pretext tasks capture different self-supervised signals from the graph. For example, link prediction tasks are more concerned with the connectivity or relationship between nodes [40], node/edge feature-based tasks focus more on the feature space [42], and subgraph-based tasks focus more on local or global information [29, 53]. To cater to diverse downstream tasks, the pre-training step should aim to broadly extract knowledge from various aspects, such as node connectivity, node or edge features, and local or global graph patterns. Hence, it is ideal to incorporate multiple pretext tasks in pre-training in order to cover a comprehensive range of knowledge.

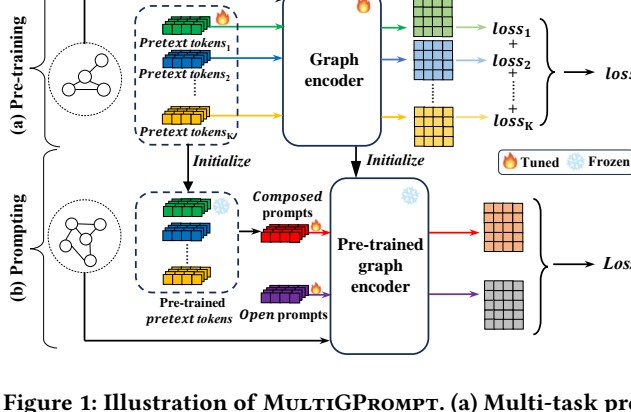

**Figure 1: Illustration of MULTIGPROMPT. (a) Multi-task pre-training on graphs. (b) Prompting on downstream tasks.**

**Challenges.** To address the limitation of a single pretext task, in this work we study multi-task pre-training for graphs and the effective transfer of multi-task knowledge to downstream few-shot tasks. The problem is non-trivial due to the following two challenges.

First, in the pre-training stage, how can we *leverage diverse pretext tasks for graph models in a synergistic manner?* A straightforward way is to sum the losses of multiple pretext tasks directly, which have been explored in language [6, 38, 48], vision [14, 32] and graph data [10, 16, 18, 30]. However, several works on multi-task learning [46, 49, 56] observe frequent task interference when tasks are highly diverse, resulting in suboptimal pre-training. A recent work [46] sheds light on more synergistic multi-task pre-training in language models via an improved transformer design, which uses an attention mask and weighted skip connection to reduce task interference. Nonetheless, it largely remains an open problem for graph models.

Second, in the adaptation stage, how can we *transfer both task-specific and global pre-trained knowledge to downstream tasks?* Multiple pretext tasks further complicates the alignment of downstream objectives with the pre-trained model. On one hand, recent prompt-based learning on graphs [11, 29, 40] only focus on the downstream adaptation to a single pretext task. On the other hand, for language models a prompt-aware attention module has been incorporated into the transformer architecture [46] to focus on extracting task-specific information from pre-training, lacking a global view of the pre-trained knowledge.

**Contributions.** To address the above two challenges, we propose a novel framework called MULTIGPROMPT for multi-task pre-training and prompting for few-shot learning on graphs.

Toward the first challenge, we draw inspiration from multi-task prompt-aware language models [46], and design a series of *pretext tokens* to synergize the pretext tasks during pre-training. As illustrated in Fig. 1(a), we associate each pretext task with one (or more) task-specific pretext tokens, which are used to reformulate the input of the pretext task. A pretext token is simply a learnable vector, and thus yields a learnable reformulation in a task-specific fashion. The reformulation guides multiple pretext tasks into a synergistic integration that enables collaboration, rather than interference, among the diverse tasks.

Towards the second challenge, we propose a dual-prompt mechanism, employing both a *composed* prompt and an *open* prompt to harness task-specific and global pre-training knowledge, respectively, as shown in Fig. 1(b). More specifically, a composed prompt is a learnable composition (e.g., a linear or neural network-based combination) of the pre-trained (frozen) pretext tokens, similar to the approach in language models [46]. As the composed prompt builds upon the pretext tokens, it is designed to query the pre-trained model for a precise mixture of information specific to each pretext task, focusing on task-specific pre-trained knowledge. However, it falls short of a global view to extract relevant inter-task knowledge (e.g., the relations or interactions between pretext tasks) from the whole pre-trained model. Hence, we propose the concept of open prompt, which aims to transfer global inter-task knowledge to complement the task-specific knowledge of the composed prompt.

To summarize, we make the following contributions in this work. (1) To the best of our knowledge, for the first time we propose MULTIGPROMPT, a multi-task pre-training and prompting framework for few-shot learning on graphs. (2) In pre-training, we introduce pretext tokens to reduce task interference, optimizing multiple pretext tasks in synergy. (3) In downstream adaptation, we propose a dual-prompt design with a composed prompt to extract task-specific pre-trained knowledge, as well as an open prompt to extract global inter-task knowledge. (4) We conduct extensive experiments on six public datasets, and the results demonstrate the superior performance of MULTIGPROMPT in comparison to the state-of-the-art approaches.

## 2 RELATED WORK

**Graph pre-training.** Borrowing insights from the realm of pre-training methodologies in both the language [5, 7, 13, 36] and vision [2, 58, 61, 62] domains, a myriad of GNN-based pre-training approaches have emerged [16, 18, 23, 30, 33]. These methods leverage the intrinsic graph structures in a self-supervised manner, setting the stage for knowledge transfer to downstream tasks. This transfer can be accomplished by a fine-tuning process that capitalizes on labeled data pertinent to each downstream task.

However, a gap emerges between the objectives of pre-training and fine-tuning [27]. On one hand, pre-training seeks to distill general knowledge from the graph without relying on explicit supervision. Conversely, fine-tuning tailors to specific supervisory signals aligned with the downstream tasks. This gap in objectives can hinder the transfer of knowledge from the pre-trained model, potentially hurting the downstream performance.

**Graph prompt learning.** Originated in the language domain [5, 47], prompt-based learning has been effective in bridging the gap between pre-training and downstream objectives. Specifically, prompts can be tuned for each downstream task, steering each task toward the pre-trained model while keeping the pre-trained parameters frozen. Due to the parameter-efficient nature of prompt, it has been quickly popularized in favor of fine-tuning larger pre-trained models, or when the downstream task only has few-shot labels. Given the advantages, prompt-based learning has also been explored on visual data [20, 21, 62] and graphs [11, 29, 40–42].

Specific to graph data, GraphPrompt [29] and ProG [41] attempt to unify pre-training and typical downstream tasks on graphs into

a common template, and further introduce learnable prompts to guide the downstream tasks. Their difference mainly lies in learning scenarios: ProG requires a set of base classes in a typical meta-learning setup [28, 44, 60], while GraphPrompt does not make use of base classes. In contrast, GPPT [40] and VNT [42] only focus on the node classification task downstream. However, all these methods only employ a single pretext task, and thus lack a comprehensive coverage of pre-trained knowledge in different aspects.

**Multi-task pre-training.** To broaden and diversify beyond a single pretext task, multi-task pre-training methods have been proposed for language [6, 38, 48], vision [14, 32] and graph data [10, 16, 18, 30]. However, on one hand, these methods directly aggregate multiple losses from diverse pretext tasks, resulting in task interference in the pre-training stage [46, 49, 56]. On the other hand, these approaches only perform fine-tuning for the downstream tasks in the adaptation stage, which is inadequate to align the multiple pretext tasks with the downstream objective. To mitigate these issues, a recent study [46] employs prompts to integrate multiple pretext tasks and further guide downstream tasks. However, it is designed for language models, requiring specific modification to the transfomer architecture. Moreover, it lacks a global view over the multiple pretext tasks.

On another line, there is some research on multi-modal prompts [8, 22], employing multiple prompts to different modality of data such as vision and language. They aim to align the representations from different modalities, which diverges from our multi-task objectives in pre-training.

## 3 PRELIMINARIES

In this work, our goal is to pre-train a graph encoder through self-supervised pretext tasks. Subsequently, the pre-trained encoder can be used for few-shot downstream tasks on graph data. We introduce related concepts and definitions in the following.

**Graph.** A graph is represented as $G = (V, E)$, with $V$ denoting the set of nodes and $E$ the set of edges. Equivalently, the graph can be represented by an adjacency matrix $\mathbf{A}$, such as $\mathbf{A}_{ij} = 1$ iff $(v_i, v_j) \in E$, for any $v_i, v_j \in V$. We further consider an input feature matrix for the nodes, given by $\mathbf{X} \in \mathbb{R}^{|V| \times d}$. For a node $v_i \in V$, its feature vector is represented as $\mathbf{x}_i \in \mathbb{R}^d$.

For a dataset with multiple graphs, we use the notation $\mathcal{G} = \{G_1, G_2, \ldots, G_N\}$.

**Graph encoder.** GNNs are popular choices of graph encoder, most of which employ a message-passing mechanism [50]. Specifically, each node in the graph aggregates messages (i.e., input features or embeddings) from its neighboring nodes to update its own embedding. Multiple layers of neighborhood aggregation can be stacked, facilitating recursive message passing across the graph. Formally, let $\mathbf{H}^l$ be the embedding matrix of the graph at the $l$-th layer, where its $i$-th row, $\mathbf{h}_i^l$, corresponds to the embedding of node $v_i$. It is computed based on the embeddings from the preceding layer:

$$\mathbf{H}^l = \text{MP}(\mathbf{H}^{l-1}, \mathbf{A}; \theta^l), \tag{1}$$

where MP$(\cdot)$ is a message passing function, $\theta^l$ denotes the learnable parameters of the graph encoder at the $l$-th layer. In particular, the initial embedding matrix $\mathbf{H}^0$, is simply given by the input feature matrix, i.e., $\mathbf{H}^0 = \mathbf{X}$. The output after a total of $L$ layers is then $\mathbf{H}^L$;

for brevity we simply write $\mathbf{H}$. We abstract the multi-layer encoding process as

$$\mathbf{H} = \text{GraphEncoder}(\mathbf{X}, \mathbf{A}; \Theta), \tag{2}$$

where $\Theta = (\theta^1, \ldots, \theta^L)$ is the collection of weights across the layers.

The output embedding matrix $\mathbf{H}$ can then be fed into a loss function and optimized. In the pre-training stage, the loss can be defined with various self-supervised pretext tasks, such as DGI [43], GraphCL [55] and link prediction [29]. In the downstream adaptation stage, the loss is computed based on labeled data.

**Few-shot problem.** For downstream tasks, we focus on few-shot learning for two graph-based tasks: node classification and graph classification. Specifically, for node classification within a graph $G = (V, E)$, let $C$ denote the set of node classes. For each node $v_i \in V$, its class label is $\ell_i \in C$. For graph classification over a set of graphs $\mathcal{G}$, we introduce $C$ as the set of possible graph labels. Here, $L_i \in C$ represents the class label for a specific graph $G_i \in \mathcal{G}$.

Under the few-shot setting, in either node or graph classification, there are only $m$ labeled examples (be it nodes or graphs) for each class, where $m$ is a small number (e.g., $m \leq 10$). This setup is commonly referred to as $m$-shot classification.

## 4 PROPOSED APPROACH

In this section, we present our proposed model MultiGPrompt.

### 4.1 Overall Framework

We begin with the overall framework of MultiGPrompt in Fig. 2, which consists of two high-level stages: (a) multi-task pre-training on some label-free graphs, and (b)/(c) prompt-based learning for few-shot downstream tasks.

First, as shown in Fig. 2(a), our framework incorporates $K$ pretext tasks for multi-task pre-training. For the $k$-th pretext task, we employ a series of pretext tokens $\mathcal{T}_{[k]}$ to store task-specific information. The pretext tokens are learnable vectors that reformulate the input of the pretext task, which guide various pretext tasks into a synergistic integration to alleviate task interference.

Next, as shown in Fig. 2(b), we aim to transfer the pre-trained knowledge to different downstream tasks. We propose a dual-prompt mechanism with a series of composed prompts, $\mathcal{P}_{\langle\text{com}\rangle}$, and open prompts, $\mathcal{P}_{\langle\text{op}\rangle}$. The composed prompt is obtained by a learnable aggregation of the pretext tokens to leverage task-specific pre-trained knowledge. The open prompt, on the other hand, is a learnable vector that learns global inter-task insights from the pre-trained model. Both are then used to reformulate the downstream input to the pre-trained model separately, and their respective output embeddings are combined and further fed into the downstream task loss.

### 4.2 Multi-task Pre-training

In this part, we discuss the first stage on multi-task pre-training. In general, any graph-based pretext tasks can be used in our framework. Without loss of generality, in our experiments, we leverage there well-known pretext tasks, namely, DGI [43], GraphCL [55], and link prediction [29]. We aim to aggregate the losses of multiple pretext tasks in a synergistic manner under the guidance of pretext tokens.

Figure 2: Overall framework of MULTIGPROMPT, consisting of two main stages: (a) Multi-task pre-training, and (b)/(c) Prompt-based learning for downstream few-shot tasks.

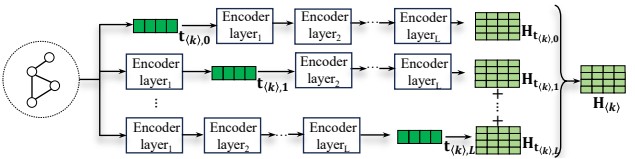

Figure 3: Application of pretext tokens to the graph encoder. $t_{\langle k \rangle, l}$ represents the pretext token that modifies the $l$-th layer of the graph encoder for the $k$-th pretext task.

**Pretext tokens.** Assume we employ a total of $K$ pretext tasks for multi-task pre-training. Different pretext tasks focus on diverse aspects, and each has its unique loss function. Directly optimizing the sum of the $K$ losses leads to interference between tasks [46, 49, 56], and degrades the pre-training efficacy.

To avoid task interference, we leverage the concept of pretext tokens, which have been used to reformulate the task input in an earlier approach for pre-training language models [46]. In the context of graph, different layers of the graph encoder may have different emphasis on the representation, and thus carry variable significance to different pretext tasks. For instance, the input layer focuses on individual nodes' features and thus are more important to node-level pretext tasks, while the hidden or output layers focus more on subgraph or graph features and thus are more important to local or graph-level tasks. Hence, we introduce a series of pretext tokens for each pretext task, to modify the input, hidden and output layers of the graph encoder alike.

Specifically, consider a graph $G$, an encoder with a total of $L$ layers, and $K$ pretext tasks. As shown in Fig. 2(a), we put forth $K$ sets of pretext tokens, represented by $\mathcal{T}_{\langle 1 \rangle}, \mathcal{T}_{\langle 2 \rangle}, \ldots, \mathcal{T}_{\langle K \rangle}$. Each $\mathcal{T}_{\langle k \rangle}$ denotes a set of $L + 1$ pretext tokens for the $k$-th pretext task, with one pretext token for each layer (including the input layer):

$$\mathcal{T}_{\langle k \rangle} = \{ t_{\langle k \rangle, 0}, t_{\langle k \rangle, 1}, \ldots, t_{\langle k \rangle, L} \}. \tag{3}$$

That is, $t_{\langle k \rangle, l}$ is a learnable vector representing the pretext token that modifies the $l$-th layer of the encoder for the $k$-th pretext task, for $1 \le k \le K$ and $0 \le l \le L$. This gives a total of $K \times (L + 1)$ pretext tokens, and we illustrate how they are applied to modify different layers for one pretext task in Fig. 3.

Next, given any pretext token $t$ in general, let $H_t$ denote the output from the graph encoder after applying the pretext token $t$

to one of its layers, as follows.

$$H_t = \text{GRAPHENCODER}_t(X, A; \Theta), \tag{4}$$

where $\text{GRAPHENCODER}_t(\cdot)$ indicate that one of its layers has been modified by $t$. To be more specific, a pretext token $t_{\langle k \rangle, l}$ will modify the $l$-th layer of the graph encoder into $t_{\langle k \rangle, l} \odot H^l$ with an element-wise multiplication, where we multiply the pretext token $t_{\langle k \rangle, l}$ with each row of $H^l$ element-wise[2]. Subsequently, when $l < L$, the next layer will be generated as

$$H^{l+1} = \text{MP}(t_{\langle k \rangle, l} \odot H^l, A; \theta^l). \tag{5}$$

Finally, for the $k$-th pretext task, we must generate one embedding matrix $H_{\langle k \rangle}$ to calculate the task loss. However, with Eq. (4), each of the $L + 1$ pretext tokens for the pretext task will generate its own embedding matrix. Thus, we further aggregate the $L + 1$ embedding matrices to obtain the overall embedding matrix for the $k$-th task as

$$H_{\langle k \rangle} = \sum_{l=0}^{L} \alpha_l H_{t_{\langle k \rangle, l}}, \tag{6}$$

where $\{\alpha_l : 0 \le l \le L\}$ are hyperparameters. Typically, a graph encoder adopts a shallow architecture with few layers. In our implementation, we have $L = 1$ and thus only two $\alpha_l$'s. Furthermore, as $\alpha_l$'s adjust the relative weights across layers, one of them can be fixed to 1, which means there is effectively only one free hyperparameter here.

**Pre-training loss.** Equipped with a set of tailored pretext tokens for each pretext task, our multi-task pre-training can capture specific information pertinent to every pretext task in synergy. After obtaining the embedding matrix for the $k$-th pretext task in Eq. (6), we can calculate the corresponding task loss $\mathcal{L}_{\text{pre}_{\langle k \rangle}}(H_{\langle k \rangle}; \mathcal{T}_{\langle k \rangle}, \Theta)$, where $\Theta$ represents the model weights of graph encoder. Note that $H_{\langle k \rangle}$ is used to calculate the loss while $\mathcal{T}_{\langle k \rangle}, \Theta$ are trainable parameters. Then, we aggregate the losses of all $K$ pretext tasks together into an overall loss for the multi-task pre-training stage:

$$\mathcal{L}_{\text{pre}}(\mathcal{H}; \mathcal{T}, \Theta) = \sum_{k=1}^{K} \beta_k \mathcal{L}_{\text{pre}_{\langle k \rangle}}(H_{\langle k \rangle}; \mathcal{T}_{\langle k \rangle}, \Theta), \tag{7}$$

---

[2]Hence, a pretext token must adopt the same dimensions as the layer it applies to.

where $\{\beta_k : 1 \leq k \leq K\}$ contains $K$ hyperparameters, $\mathcal{H} = \{\mathbf{H}_{\langle 1 \rangle}, \ldots, \mathbf{H}_{\langle K \rangle}\}$ represents the collection of task-specific embeddings, and $\mathcal{T} = \{\mathcal{T}_{\langle 1 \rangle}, \ldots, \mathcal{T}_{\langle K \rangle}\}$ denotes the collection of pretext token sets. The overall loss is optimized by updating the pretext tokens $\mathcal{T}$ and encoder weights $\Theta$. Note that the number of pretext tasks $K$ should be a small constant, with $K = 3$ in our experiments. Similar to $\alpha_l$'s, one of $\beta_k$'s can be fixed at 1, leaving only two free hyperparameters.

Specifically, we employ DGI [43], GraphCL [55], and Link Prediction [29] as pretext tasks. Additional details are provided in Appendix D.

## 4.3 Prompting for Downstream Tasks

To leverage not only task-specific pre-trained knowledge, but also global inter-task knowledge from the whole pre-trained model, we propose a dual-prompt mechanism with a set of composed prompts, $\mathcal{P}_{\langle \text{com} \rangle}$, and a set of open prompts, $\mathcal{P}_{\langle \text{op} \rangle}$. Composed prompts aim at transferring pretext task-specific knowledge to downstream tasks, through a learnable mixture of pretext tokens. Simultaneously, open prompts facilitate the transfer of global inter-task knowledge.

Both composed prompts and open prompts are applied to different layers of the pre-trained graph encoder in the same manner as pretext tokens, as illustrated in Fig. 3. That is, the set of composed prompts $\mathcal{P}_{\langle \text{com} \rangle}$ contains $L + 1$ prompts, so does the set of open prompts $\mathcal{P}_{\langle \text{op} \rangle}$, as follows.

$$\mathcal{P}_{\langle \text{com} \rangle} = \{\mathbf{p}_{\langle \text{com} \rangle, 0}, \mathbf{p}_{\langle \text{com} \rangle, 1}, \ldots, \mathbf{p}_{\langle \text{com} \rangle, L}\} \quad (8)$$

$$\mathcal{P}_{\langle \text{op} \rangle} = \{\mathbf{p}_{\langle \text{op} \rangle, 0}, \mathbf{p}_{\langle \text{op} \rangle, 1}, \ldots, \mathbf{p}_{\langle \text{op} \rangle, L}\}, \quad (9)$$

Each prompt $\mathbf{p} \in \mathcal{P}_{\langle \text{com} \rangle}$ or $\mathcal{P}_{\langle \text{op} \rangle}$ is a vector that modifies a specific layer of the pre-trained encoder. Similar to Eq. (10), let $\mathbf{H}_{\mathbf{p}}$ be the output from the pre-trained graph encoder after applying the prompt $\mathbf{p}$ to one of its layers, as follows.

$$\mathbf{H}_{\mathbf{p}} = \text{GraphEncoder}_{\mathbf{p}}(\mathbf{X}, \mathbf{A}; \Theta_{\text{pre}}), \quad (10)$$

where $\Theta_{\text{pre}}$ contains pre-trained model weights that are frozen throughout the downstream stage. Then, define $\mathbf{H}_{\langle \text{com} \rangle}$ and $\mathbf{H}_{\langle \text{op} \rangle}$ as the final output of the pre-trained graph encoder after applying the composed prompts and open prompts, respectively. That is,

$$\mathbf{H}_{\langle \text{com} \rangle} = \sum_{l=0}^{L} \alpha_l \mathbf{H}_{\mathbf{p}_{\langle \text{com} \rangle, l}}, \quad \mathbf{H}_{\langle \text{op} \rangle} = \sum_{l=0}^{L} \alpha_l \mathbf{H}_{\mathbf{p}_{\langle \text{op} \rangle, l}}, \quad (11)$$

where $\alpha_l$'s take the same values as those in Eq. (6).

In the following, we elaborate how a composed prompt and an open prompt is constructed.

**Composed prompt.** As given in Eq. (8), a composed prompt $\mathbf{p}_{\langle \text{com} \rangle, l} \in \mathcal{P}_{\langle \text{com} \rangle}$ modifies the $l$-th layer of the pre-trained graph encoder, following the same fashion as Eq. (5). However, $\mathbf{p}_{\langle \text{com} \rangle, l}$ is not directly learnable, but is instead a learnable composition of the $K$ pre-trained pretext tokens in the same layer, as given below.

$$\mathbf{p}_{\langle \text{com} \rangle, l} = \text{Compose}(\mathbf{t}_{\langle 1 \rangle, l}, \mathbf{t}_{\langle 2 \rangle, l}, \ldots, \mathbf{t}_{\langle K \rangle, l}; \Gamma), \quad (12)$$

where $\text{Compose}(\cdot)$ is a function to "compose" the $K$ pretext tokens together, such as a linear combination or neural network, and $\Gamma$ represents the learnable parameters of the function. Therefore, a composed prompt aims to learn a precise mixture of task-specific pre-trained knowledge.

## Table 1: Summary of datasets.

| | Graphs | Graph classes | Avg. nodes | Avg. edges | Node features | Node classes | Task[*] (N/G) |
|---|---|---|---|---|---|---|---|
| Cora | 1 | - | 2,708 | 5,429 | 1,433 | 7 | N |
| Citeseer | 1 | - | 3,327 | 4,732 | 3,703 | 6 | N |
| PROTEINS | 1,113 | 2 | 39.06 | 72.82 | 1 | 3 | N, G |
| ENZYMES | 600 | 6 | 32.63 | 62.14 | 18 | 3 | N, G |
| BZR | 405 | 2 | 35.75 | 38.36 | 3 | - | G |
| COX2 | 467 | 2 | 41.22 | 43.45 | 3 | - | G |

[*] indicates the type(s) of downstream task associated with each dataset: "N" for node classification and "G" for graph classification.

**Open prompt.** Similar to a composed prompt, an open prompt $\mathbf{p}_{\langle \text{op} \rangle, l} \in \mathcal{P}_{\langle \text{op} \rangle}$ modifies the $l$-th layer of the pre-trained graph encoder. However, unlike the composed prompts, $\mathbf{p}_{\langle \text{op} \rangle, l}$ is directly tuned, instead of being composed from the pretext tokens. In this way, an open prompt does not extract pre-trained knowledge specific to any pretext task, but focus on the global pre-trained model holistically.

**Prompt tuning.** Lastly, we generate a final embedding matrix to compute the downstream task loss. To leverage not only pretext task-specific knowledge but also global information from the pre-trained model, we incorporate the output embeddings from both the composed and open prompts given by Eq. (11). To this end, let us define an aggregation function $\text{Aggr}(\cdot)$, which gives the final embedding matrix $\tilde{\mathbf{H}}$ after applying the dual prompts to the pre-trained encoder, as follows.

$$\tilde{\mathbf{H}} = \text{Aggr}(\mathbf{H}_{\langle \text{com} \rangle}, \mathbf{H}_{\langle \text{op} \rangle}; \Delta), \quad (13)$$

where $\Delta$ denotes the set of learnable parameters of the aggregation function.

To tune the dual prompts for an arbitrary downstream task, the loss can be abstracted as $\mathcal{L}_{\text{down}}(\tilde{\mathbf{H}}; \mathcal{P}_{\langle \text{op} \rangle}, \Gamma, \Delta)$, where $\tilde{\mathbf{H}}$ is used to calculate the loss, and $\mathcal{P}_{\langle \text{op} \rangle}, \Gamma, \Delta$ are tunable parameters associated with the prompts. Note that during prompt tuning, the pre-trained weights of the graph encoder and the pretext tokens are frozen without any tuning. Only $\mathcal{P}_{\langle \text{op} \rangle}, \Gamma, \Delta$ are updated, which is much more parameter-efficient than fine-tuning the pre-trained model. Hence, our approach is particularly suitable for few-shot settings when the downstream task only offers a few labeled examples.

More concretely, in this work, we have experimented with two popular types of downstream task, namely, node classification and graph classification, and follow the same loss formulations in a previous work [29]. Details of the losses can be found in Appendix E.

## 5 EXPERIMENTS

In this section, we undertake comprehensive experiments across six benchmark datasets, to evaluate the efficacy of the proposed MultiGPrompt on few-shot node classification and graph classification tasks.

## 5.1 Experimental Setup

**Datasets.** We employ six benchmark datasets for evaluation. (1) *Cora* [31] and (2) *Citeseer* [37] are both citation graphs. Each of them involves a single graph, where the nodes are publications and the edges are citations. As with previous work [24, 43], we

**Table 2: Accuracy evaluation on few-shot node and graph classification.**

| Methods | Node classification | | | | Graph classification | | | |
|---|---|---|---|---|---|---|---|---|
| | Cora | Citeseer | PROTEINS | ENZYMES | BZR | COX2 | PROTEINS | ENZYMES |
| GCN | 28.57 ± 5.07 | 31.27 ± 4.53 | 43.31 ± 9.35 | 48.08 ± 4.71 | 56.33 ± 10.40 | 50.95 ± 23.48 | 50.56 ± 3.01 | 17.10 ± 3.53 |
| GAT | 28.40 ± 6.25 | 30.76 ± 5.40 | 31.79 ± 20.11 | 35.32 ± 18.72 | 50.69 ± 23.66 | 50.58 ± 26.16 | 50.59 ± 12.43 | 16.80 ± 2.97 |
| DGI/InfoGraph | 54.11 ± 9.60 | 45.00 ± 9.19 | 45.22 ± 11.09 | 48.05 ± 14.83 | 52.57 ± 18.14 | 54.62 ± 15.36 | 48.21 ± 12.35 | 21.69 ± 5.98 |
| GraphCL | 51.96 ± 9.43 | 43.12 ± 9.61 | 46.15 ± 10.94 | 48.88 ± 15.98 | 54.11 ± 16.63 | 54.29 ± 17.31 | 53.69 ± 11.92 | 21.57 ± 5.20 |
| GPPT | 15.37 ± 4.51 | 21.45 ± 3.45 | 35.15 ± 11.40 | 35.37 ± 9.37 | - | - | - | - |
| GraphPrompt | 54.25 ± 9.38 | 45.34 ± 10.53 | 47.22 ± 11.05 | 53.54 ± 15.46 | 54.60 ± 10.53 | 54.35 ± 14.78 | 54.73 ± 8.87 | 25.06 ± 7.56 |
| MultiGPrompt | **57.72** ± 9.94 | **54.74** ± 11.57 | **48.09** ± 11.49 | **54.47** ± 15.36 | **60.07** ± 12.48 | **56.17** ± 12.84 | **56.02** ± 8.27 | **26.63** ± 6.22 |

Results are reported in percent. The best method is bolded and the runner-up is underlined.

treat their edges as undirected. (3) *PROTEINS* [4] comprises 1,113 protein graphs. Each node signifies a secondary structure, and the edges represent the neighboring relations between the structures, within the amino-acid sequence or in 3D space. (4) *ENZYMES* [45], (5) *BZR* [35], and (6) *COX2* [35] are collections of molecular graphs, describing the structures of 600 enzymes from the BRENDA enzyme database, 405 ligands pertinent to the benzodiazepine receptor, and 467 cyclooxygenase-2 inhibitors, respectively.

We summarize the characteristics of these datasets in Table 1, and provide a comprehensive description in Appendix A.

**Baselines.** We evaluate MultiGPrompt against a spectrum of state-of-the-art methods that can be broadly grouped into three primary categories, as follows.

(1) *End-to-end graph neural networks*: These include GCN [24] and GAT [43], which are trained in a supervised manner using the downstream labels directly, without pre-training.

(2) *Graph pre-training models*: We compare to DGI/InfoGraph[3] [39, 43], and GraphCL [55]. Specifically, the pre-training stage only uses label-free graphs, and the downstream adaptation stage further trains a classifier using few-shot labels while freezing the pre-trained weights.

(3) *Graph prompt-based learning*: GPPT [40] and GraphPrompt [29] are included under this umbrella. Their modus operandi revolves around leveraging link prediction during pre-training, and unifying downstream tasks into a common template as the pretext task. Both of them utilizes a single pretext task, and subsequently a single type of prompt in downstream adaptation. Note that GPPT is purposely designed to work with the downstream task of node classification, and cannot be directly applied to graph classification. Hence, in our experiments, we only employ GPPT for node classification.

We present more details of the baselines in Appendix B. It is worth noting that certain few-shot methodologies on graphs, such as Meta-GNN [60], AMM-GNN [44], RALE [28], VNT [42], and ProG [41], hinge on the meta-learning paradigm [12], requiring an additional set of labeled base classes in addition to the few-shot classes. Hence, they are not comparable to our framework.

**Setup of downstream tasks.** We conduct two types of downstream task, i.e., node classification and graph classification. The

tasks are configured in a $m$-shot classification setup, i.e., for each class, we randomly draw $m$ examples (nodes or graphs) as supervision. In our main results, we use $m = 1$ for node classification, and $m = 5$ for graph classification. Nevertheless, we also vary the number of shots for $1 \leq m \leq 10$, to show the robustness of our approach under different settings.

We repeat the sampling 100 times to construct 100 $m$-shot tasks for node classification as well as graph classification. For each task, we run with five different random seeds. Thus, there are a total of 500 results per type of task, and we report the average and standard deviation over the 500 results. Since the $m$-shot tasks are balanced classification, we simply evaluate the performance using accuracy, in line with previous works [28, 29, 44].

Note that, for datasets with both node and graph classification tasks, i.e., *PROTEINS* and *ENZYMES*, we only pre-train the graph encoder once for each dataset. Subsequently, we employ the same pre-trained model for both types of downstream task.

**Parameter settings.** For all baselines, we reuse the original authors' code and their recommended settings, and further tune their hyper-parameters to ensure competitive performance. A more granular description of the implementations and settings, for both the baselines and our MultiGPrompt, is furnished in Appendix C.

## 5.2 Few-shot Performance Evaluation

We first report the performance of one-shot node classification and five-shot graph classification. Next, we study the impact of varying number of shots on the performance.

**One-shot node classification.** The results are presented in Table 2. We make the following observations.

First, MultiGPrompt surpasses all baselines on all four datasets, indicating its advantage in the overall strategy of multi-task pre-training. We will further conduct a series of ablation studies in Sect. 5.3 to evaluate the importance of specific designs. Second, pre-training methods (DGI/InfoGraph, GraphCL) generally outperform supervised methods (GCN, GAT), as the former group leverage a pre-trained model. The results highlight the importance of acquiring general knowledge from label-free graphs. Lastly, "pre-train, prompt" methods, such as GraphPrompt and our MultiG-Prompt, can further outperform the pre-training approaches without prompts, demonstrating the advantage of prompt-based learning epsecially in few-shot settings.

---

[3]Original DGI only works at the node level, while InfoGraph extends it to the graph level. In our experiments, we use DGI for node classification, and InfoGraph for graph classification.

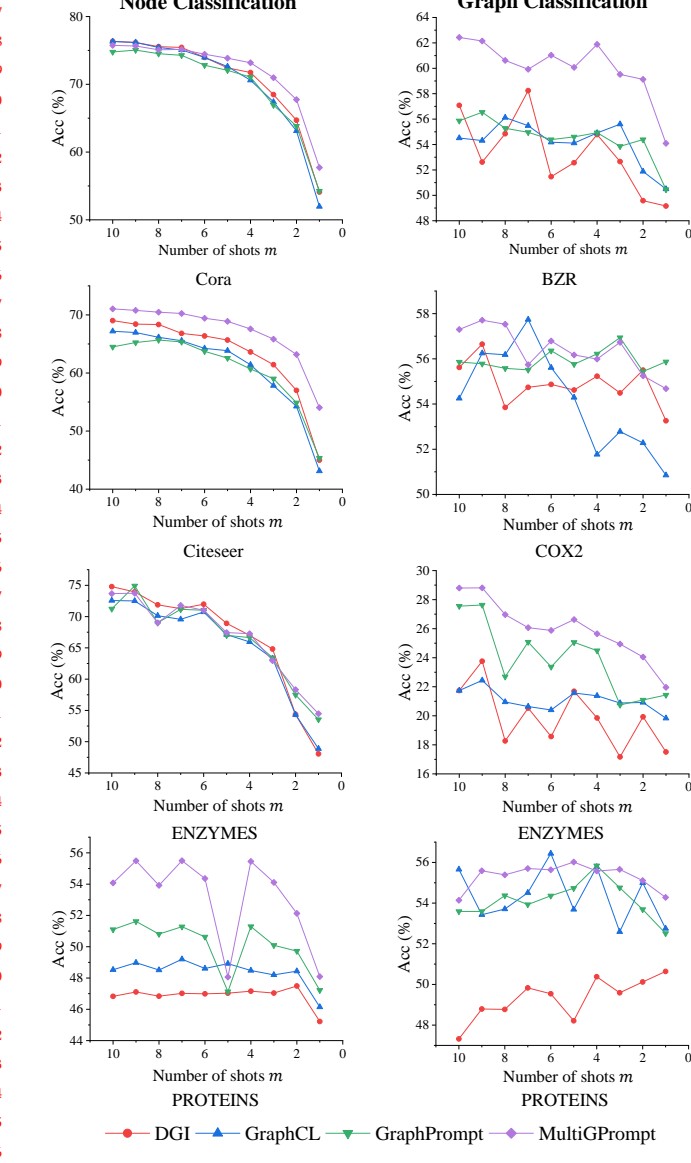

Figure 4: Impact of shots on node and graph classification.

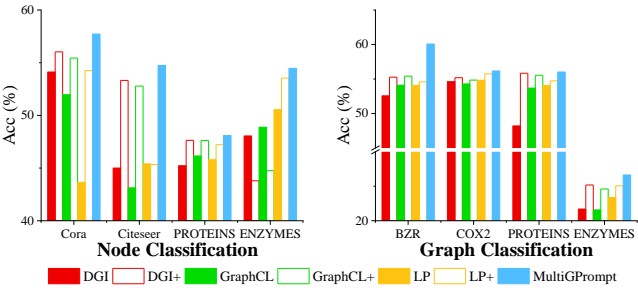

Figure 5: Ablation study on pretext tasks.

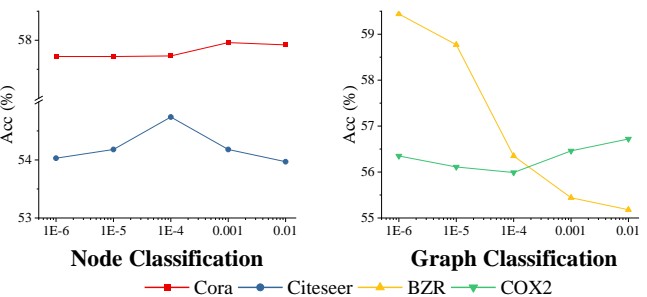

Figure 6: Impact of hyperparameter $\alpha_0$.

**Five-shot graph classification.** We further conduct graph classification and also report the results in Table 2. The trends on graph classification are mostly consistent with those observed in the node classification results, underpinning the generality of MultiG-Prompt (and more broadly, the paradigm of prompt-based learning) across both node- and graph-level tasks.

**Impact of different shots.** To delve deeper into the robustness of MultiGPrompt in different learning setups, we vary the the number of shots $m$ for both node and graph classification tasks. We present the performance of MultiGPrompt against a line-up of competitive baselines in Fig. 4, and make several observations.

First, MultiGPrompt largely performs better than the baselines, especially in low-shot settings (e.g., $m \leq 5$) when very limited labeled data are given. Second, when more shots are given (e.g.,

$m > 5$), all methods perform better in general, which is expected. Nonetheless, the performance of MultiGPrompt remains competitive, if not better. Note that on certain datasets such as *PROTEINS*, the performance of most methods suffer from large variances. One potential reason is it has least node features (see Table 1), which exacerbate the difficulty of few-shot classification. Additionally, the graphs in *PROTEINS* tend to vary in size more significantly than other datasets[4], which may also contribute to the larger variances in the performance. Despite these issues, the performance of MultiGPrompt is still more robust than other methods.

## 5.3 Ablation study

To thoroughly understand the impact of each component within MultiGPrompt, we conduct two ablation analyses. The initial analysis studies the effect of multiple pretext tasks, and the second analysis contrasts MultiGPrompt with variants employing different prompts.

We start with three basic variants that only utilize a single pretext task: using only DGI/InfoGraph (DGI), GraphCL, and link prediction (LP). These three basic variants simply employ a classifier during downstream fine-tuning, without any prompting. We further compare three more advanced variants, namely, DGI+, GraphCL+ and LP+, which has the exact same architecture and dual-prompt design as the full model MultiGPrompt, but only utilize one pretext task. Referring to Fig. 5, we observe that MultiGPrompt consistently outperforms all variants using a single pretext task, with or without prompts. This finding underscores the value of leveraging multiple pretext tasks.

---

[4]The graph sizes in *PROTEINS* has a standard deviation of 45.78, where other datasets lie in the range between 4.04 and 15.29.

**Table 3: Ablation study on prompt design for multi-task pre-training.**

| Methods | Pretext token | Composed prompt | Open prompt | Node classification | | | | Graph classification | | | |
|---|---|---|---|---|---|---|---|---|---|---|---|
| | | | | Cora | Citeseer | PROTEINS | ENZYMES | BZR | COX2 | PROTEINS | ENZYMES |
| Variant 1 | × | × | × | 56.58 | 50.69 | 46.48 | 48.04 | 49.63 | 54.35 | 55.72 | 21.07 |
| Variant 2 | × | × | ✓ | 56.54 | 53.08 | 47.79 | 51.09 | 47.56 | 54.89 | 55.61 | 24.23 |
| Variant 3 | ✓ | × | × | 45.00 | 52.36 | 45.11 | 50.55 | 57.14 | 54.43 | 55.67 | 21.06 |
| Variant 4 | ✓ | × | ✓ | 56.59 | 50.63 | 47.64 | 50.52 | 57.52 | 55.21 | 55.12 | 24.30 |
| Variant 5 | ✓ | ✓ | × | 56.83 | 53.72 | 47.50 | 53.11 | 55.71 | 53.04 | 55.15 | 23.33 |
| MultiGPrompt | ✓ | ✓ | ✓ | **57.72** | **54.74** | **48.09** | **54.47** | **60.07** | **56.17** | **56.02** | **26.63** |

Results are evaluated using classification accuracy, reported in percent. The best variant is bolded.

Next, for multi-task pre-training, we investigate several variants of MultiGPrompt by removing key designs in our dual prompts, including the use of pretext tokens, composed prompts and open prompts. These variants and their corresponding results are tabulated in Table 3. The outcomes corroborate that each individual design is instrumental, as analyzed below. First, employing pretext tokens and composed prompts is beneficial. Notably, Variant 5 typically outperforms Variants 1 and 3, which do not utilize a composed prompt. However, solely employing pretext tokens, as in Variant 3, does not give a stable improvement over Variant 1, implying that the pretext tokens work the best in conjunction with the composed prompts. (Note that composed prompts are built upon the pretext tokens and cannot work alone without the latter.) Second, omitting open prompts leads to diminished performance, as evident in the higher accuracy of Variants 2 and 4 against Variants 1 and 3. This shows the importance of leveraging global inter-task knowledge via open prompts. Lastly, the dual-prompt design, comprising both composed and open prompts, proves beneficial, helping MultiGPrompt achieve the most optimal performance.

## 5.4 Further Model Analysis

We conduct further analysis related to the hyperparameter selection and parameter efficiency of MultiGPrompt.

**Hyperparameter selection.** We assess the impact of $\alpha^l$'s, used in Eqs. (6) and (11), which adjust the balance between the pretext tokens or prompts across different layers. In our experiments, we only employ one message-passing layer in the graph encoder, i.e., $L = 1$, giving us two hyperparameters $\alpha_0$ and $\alpha_1$. Note that $\alpha_0$ controls the importance of prompts on the input layer, whereas $\alpha_1$ controls that of the output layer of the graph encoder. We fix $\alpha_1 = 1$ while varying $\alpha_0$, and illustrate its impact in Fig. 6.

We observe that, for node classification, as $\alpha_0$ increases, the accuracy initially rises. Upon reaching a peak, the accuracy begins to gradually decline with further increases in $\alpha_0$. In contrast, for graph classification, it appears to follow a different trend. As $\alpha_0$ decreases, accuracy tends to improve. Overall, node classification tasks favor somewhat larger $\alpha_0$, while graph classification tasks lean toward smaller $\alpha_0$. This phenomenon could be attributed to the differences in node classification and graph classification. Being a node-level task, node classification naturally focuses more on the node's input features, while graph classification is affected more by the global graph representation. Hence, node classification would place a higher weight on the input layer as controlled by $\alpha_0$, whereas graph classification would weigh the output layer more,

**Table 4: Comparison of the number of tunable parameters during the downstream adaptation stage.**

| Methods | Node classification | | Graph classification | |
|---|---|---|---|---|
| | Cora | Citeseer | BZR | COX2 |
| GCN | 368,640 | 949,504 | 1,280 | 1,280 |
| DGI/InfoGraph | 1,792 | 1,536 | 768 | 768 |
| GraphCL | 1,792 | 1,536 | 768 | 768 |
| GraphPrompt | 256 | 256 | 256 | 256 |
| MultiGPrompt | 522 | 522 | 522 | 522 |

which captures comparatively more global information across the graph after undergoing the message-passing layers.

**Parameters efficiency.** Lastly, we analyze the parameter efficiency of our approach MultiGPrompt in comparison to other representative methods. Specifically, we calculate the number of parameters that require updating or tuning during the downstream adaptation stage, and list the statistics in Table 4. For GCN, as it is trained end-to-end, all the model weights have to be updated, leading to the worst parameter efficiency. For DGI/InfoGraph and GraphCL, we only update the downstream classifier without updating the pre-trained model, resulting in a significant reduction in the number of tunable parameters. Finally, prompt-based methods GraphPrompt and MultiGPrompt are the most parameter efficient, as prompts are light-weight and contain fewer parameters than a typical classifier such as a fully connected layer. Note that, due to our dual-prompt design, MultiGPrompt needs to update more parameters than GraphPrompt in the downstream adaptation stage. However, the increase in the tunable parameters downstream is still insignificant compared to updating the classifier or the full model weights, and thus does not present a fundamental problem.

## 6 CONCLUSIONS

In this paper, we explored multi-task pre-training and prompting on graphs, aiming to encompass a comprehensive range of knowledge from diverse pretext tasks. Our proposed approach MultiGPrompt designs a series of pretext tokens to leverage multiple pretext tasks in a synergistic manner. Moreover, we introduced a dual-prompt mechanism with both composed prompts and open prompts, to utilize both pretext task-specific and global inter-task knowledge. Finally, we conducted extensive experiments on six public datasets and demonstrated that MultiGPrompt significantly outperforms various state-of-the-art baselines.

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

# APPENDICES

## A    Further Descriptions of Datasets

We provide further details of the datasets.

(1) *Cora*[5] [31] comprises 2,708 scientific publications, each categorized into one of seven classes. The citation network encompasses 5,429 links. Every publication in the dataset is depicted by a 0/1-valued word vector, indicating the absence/presence of the corresponding word from the dictionary, which consists of 1433 unique words.

(2) *Citeseer*[6] [37] is composed of 3,312 scientific publications, each categorized into one of six classes. The citation network entails 4,732 links. Each publication in the dataset is characterized by a 0/1-valued word vector, indicating the absence/presence of the corresponding word from the dictionary, which encompasses 3,703 unique words.

(3) *PROTEINS*[7] [4] encompasses a collection of protein graphs, embodying various attributes including the amino acid sequence, conformation, structure, and distinctive features such as active sites of the proteins. In this collection, each node signifies the secondary structures, whereas each edge represents the neighboring relation either within the amino-acid sequence or in 3D space. The nodes are classified into three categories, while the graphs are divided into two classes.

(4) *ENZYMES*[8] [45] constitutes a dataset comprising 600 enzymes, meticulously collected from the BRENDA enzyme database. These enzymes are meticulously categorized into 6 distinct classes in accordance with their top-level EC enzyme classification.

(5) *BZR*[9] [35] encompasses a collection of 405 ligands, each associated with the benzodiazepine receptor and graphically represented as individual entities. The entire ligand set is bifurcated into 2 distinct categories.

(6) *COX2*[10] [35] encompasses a dataset that includes 467 molecular structures, specifically of cyclooxygenase-2 inhibitors, wherein each node symbolizes an atom and each edge represents the chemical bond—be it single, double, triple, or aromatic—between atoms. The entirety of the molecules is categorized into two classes.

We conduct node classification on *Cora*, *Citeseer*, *PROTEINS*, and *ENZYMES*, by aggregating the graphs within a dataset into a larger graph. Additionally, graph classification is conducted on *PROTEINS*, *COX2*, *ENZYMES*, and *BZR*.

## B    Further Descriptions of Baselines

In this section, we present more details for the baselines.

### (1) End-to-end Graph Neural Networks

- **GCN** [24]: GCN employs a mean-pooling-based neighborhood aggregation approach to consolidate messages from adjacent nodes.
- **GAT** [43]: GAT also leverages neighborhood aggregation for end-to-end node representation learning, distinguishes

itself by allocating varied weights to neighboring nodes, thus modifying their influence in the aggregation process.

### (2) Graph Pre-training Models

- **DGI** [43]: DGI operates as a self-supervised pre-training methodology tailored for homogeneous graphs. It is predicated on the maximization of mutual information (MI), aiming to enhance the estimated MI between locally augmented instances and their global counterparts.
- **InfoGraph** [39]: Expanding upon DGI, InfoGraph is centered on graph-level tasks, endeavoring to maximize the alignment between node and graph embeddings.
- **GraphCL** [55]: GraphCL leverages a variety of graph augmentations for self-supervised learning, tapping into the intrinsic structural patterns of graphs. The overarching goal is to amplify the concordance between different augmentations throughout graph pre-training.

### (3) Graph Prompt Models

- **GPPT** [40]: GPPT embraces a GNN model, pre-trained through executing a link prediction task. The utilization of a prompt module structures the downstream node classification task, orchestrating it cohesively with the link prediction format.
- **GraphPrompt** [29]: GraphPrompt employs subgraph similarity calculations as a mechanism to amalgamate pre-training and downstream tasks, inclusive of node and graph classification. A learnable prompt is subsequently refined during the execution of the downstream task to incorporate task-specific nuances.

## C    Implementation Details of Approaches

**Details of baselines.** For all the baseline models, we utilize the codes officially disseminated by their respective authors. Each model is tuned in accordance with the settings recommended in their respective literature to ascertain optimal performance.

For the baseline GCN [24], we employ a 3-layer architecture, and set the hidden dimensions to 256. For GAT [43], we employ a 2-layer architecture and set the hidden dimension to 64. Additionally, we apply 8 attention heads in the first GAT layer.

For DGI [43], we utilize a 1-layer GCN as the base model and set the hidden dimension to 256. Additionally, we employ prelu as the activation function. For GraphCL [55], a 1-layer GCN is also employed as its base model, with the hidden dimensions set to 256. Specifically, we select edge dropping as the augmentations, with a default augmentation ratio of 0.2.

For GPPT [40], we utilize a 2-layer GraphSAGE as its base model, setting the hidden dimension to 256. For base GraphSAGE, we also employ a mean aggregator. For GraphPrompt [29], a 3-layer GCN is used as the base model for all datasets, with the hidden dimensions set to 256.

**Details of MULTIGPROMPT.** For our proposed MULTIGPROMPT, we utilize a 1-layer GCN as the base model for all datasets, assigning the hidden dimensions a value of 256. We designate $\alpha_0 = 0.0001$ for node classification tasks, while setting $\alpha_0 = 0$ for graph classification tasks. And $\alpha_1$ is set to 1. The parameters $\beta_1$, $\beta_2$, and $\beta_3$ are set to 0.9, 0.9, and 0.1 respectively.

---

[5]https://relational.fit.cvut.cz/dataset/CORA
[6]https://nrvis.com/download/data/labeled/citeseer.zip
[7]https://www.chrsmrrs.com/graphkerneldatasets/PROTEINS.zip
[8]http://www.chrsmrrs.com/graphkerneldatasets/ENZYMES.zip
[9]https://www.chrsmrrs.com/graphkerneldatasets/BZR.zip
[10]https://www.chrsmrrs.com/graphkerneldatasets/COX2.zip

## D Pretext Tasks

In our experiments, we employ three novel pretext tasks, *i.e.*, DGI [43], GraphCL [55] and link prediction [29]. Consequently, given a pretext task $k$ and graph $G$. $\mathcal{T}_{\langle k \rangle}$ serves as the pretext tokens for pretext task $k$. $\mathcal{A}$ denotes the set of positive samples, $\mathcal{B}$ represents the set of corresponding negative samples. Define $\mathbf{h}_{\langle k \rangle, v}$, a row of $\mathbf{H}_{\langle k \rangle}$, as node $v$'s representation. $\mathbf{h}_{\langle k \rangle, G}$ denotes the representation of $G$, that is:

$$\mathbf{h}_{\langle k \rangle, G} = \text{READOUT}(\mathbf{H}_{\langle k \rangle}). \tag{14}$$

Therefore, DGI's loss function $\mathcal{L}_{\text{pre}_{\langle \text{DGI} \rangle}}(\mathbf{H}_{\langle \text{DGI} \rangle}; \mathcal{T}_{\langle \text{DGI} \rangle}, \Theta) =$

$$\frac{1}{|\mathcal{A}| + |\mathcal{B}|} \Big( \sum_{a \in \mathcal{A}} \log[\text{sim}(\mathbf{h}_{\langle \text{DGI} \rangle, a}, \mathbf{h}_{\langle \text{DGI} \rangle, G})] + \\ \sum_{b \in \mathcal{B}} \log[1 - \text{sim}(\mathbf{h}_{\langle \text{DGI} \rangle, b}, \mathbf{h}_{\langle \text{DGI} \rangle, G})] \Big), \tag{15}$$

where $|\mathcal{A}|$ represents the number of positive samples, similarly $|\mathcal{B}|$ serves as the number of negative samples.

Similarly, the pre-training loss for GraphCL (hereinafter abbreviated as GCL) $\mathcal{L}_{\text{pre}_{\langle \text{GCL} \rangle}}(\mathbf{H}_{\langle \text{GCL} \rangle}; \mathcal{T}_{\langle \text{GCL} \rangle}, \Theta) =$

$$-\ln \frac{\exp(\text{sim}(\mathbf{h}_{\langle \text{GCL} \rangle, a}, \mathbf{h}_{\langle \text{GCL} \rangle, v}) / \tau)}{\sum_{b \in \mathcal{B}} \exp(\text{sim}(\mathbf{h}_{\langle \text{GCL} \rangle, b}, \mathbf{h}_{\langle \text{GCL} \rangle, v}) / \tau)}. \tag{16}$$

Finally, let the 1-hop subgraph of node $v$ be denoted as $S_v$, and the pre-training loss for link prediction (subsequently abbreviated as LP) is $\mathcal{L}_{\text{pre}_{\langle \text{LP} \rangle}}(\mathbf{H}_{\langle \text{DGI} \rangle}; \mathcal{T}_{\langle \text{LP} \rangle}, \Theta) =$

$$- \sum_{(v, a, b) \in \mathcal{X}_{\text{pre}}} \ln \frac{\exp(\text{sim}(\mathbf{h}_{\langle \text{LP} \rangle, S_v}, \mathbf{h}_{\langle \text{LP} \rangle, S_a}) / \tau)}{\sum u \in \{a, b\} \exp(\text{sim}(\mathbf{h}_{\langle \text{LP} \rangle, S_v}, \mathbf{h}_{\langle \text{LP} \rangle, S_b}) / \tau)}, \tag{17}$$

where $\mathbf{h}_{\langle \text{LP} \rangle, S_v}$ serves as the vector representation of $S_v$ with pretext tokens $\mathcal{T}_{\langle \text{LP} \rangle}$ which is calculated in the same way as Eq. (14). $\mathcal{X}_{\text{pre}}$ denotes the sampled tuples utilized for link prediction [29], $a$ serves as positive sample and $b$ denotes negative samples.

## E Prompt Tuning Loss

We resort to a loss based on node/graph similarity. Consider an NC or GC task with a labeled training set $\mathcal{D}_{\text{down}} = \{(x_1, y_1), (x_2, y_2), \ldots\}$, where $x_i$ is either a node or a graph, and $y_i \in Y$ is $x_i$'s class label from a set of classes $Y$. Then, the prompt tuning loss is defined as $\mathcal{L}_{\text{down}}(\tilde{\mathbf{H}}; \mathcal{P}_{\text{op}}, \Gamma, \Delta) =$

$$- \sum_{(x_i, y_i) \in \mathcal{D}_{\text{down}}} \ln \frac{\exp\left(\frac{1}{\tau} \text{sim}(\tilde{\mathbf{h}}_{x_i}, \overline{\tilde{\mathbf{h}}}_{y_i})\right)}{\sum_{c \in Y} \exp\left(\frac{1}{\tau} \text{sim}(\tilde{\mathbf{h}}_{x_i}, \overline{\tilde{\mathbf{h}}}_c)\right)}, \tag{18}$$

where $\tilde{\mathbf{h}}_{x_i}$ denotes the final embedding of node/graph $x_i$. Class $c$ prototype embedding $\overline{\tilde{\mathbf{h}}}_c$ is also generated based on the dual-prompt combination of composed prompt and open prompt.

Specifically, for node $v$, $\tilde{\mathbf{h}}_v$ is a row of $\tilde{\mathbf{H}}$. While for graph $G$,

$$\tilde{\mathbf{h}}_G = \text{READOUT}(\tilde{\mathbf{H}}). \tag{19}$$

## F Cross Data Scenario

To further analyze the robustness of MULTIGPROMPT, we conduct additional experiments using various datasets for pre-training and downstream prompting. Specifically, we select *BZR* and *COX2*, as

**Table 5: Accuracy evaluation for cross data scenario on graph classification.**

| Pretext | BZR | | COX2 | |
|---|---|---|---|---|
| Downstream | BZR | COX2 | BZR | COX2 |
| INFOGRAPH | 52.57 ± 18.14 | 53.71 ± 16.07 | 52.44 ± 24.49 | 54.62 ± 15.36 |
| GRAPHCL | 54.11 ± 16.63 | 51.60 ± 16.78 | 53.61 ± 24.38 | 54.29 ± 17.31 |
| GRAPHPROMPT | 54.60 ± 10.53 | 55.48 ± 11.36 | 53.78 ± 11.92 | 54.34 ± 14.78 |
| MULTIGPROMPT | **60.07** ± 12.48 | **55.81** ± 13.08 | **59.58** ± 13.03 | **56.17** ± 12.84 |

they possess the same node attribute dimensions, thus the pre-trained model can directly adapted to the other dataset. We respectively employ *BZR* and *COX2* for pre-training, and perform prompting on these two datasets. The results are delineated in Table 5. We have the following observations. First, MULTIGPROMPTconsistently surpasses other baselines even in cross-data scenario, demonstrating the robustness of the overall multi-task pre-training and prompting approach. Second, multi-task pre-training and prompting on various data do not necessarily lead to diminished performance compared to operations on identical data. This further attests to the efficacy of multi-task pre-training in acquiring effective pretext knowledge and transferring it effectively via a dual-prompt approach.

