# OpenReview forum: "MultiGPrompt for Multi-Task Pre-Training and Prompting on Graphs"
_ACM.org/TheWebConf/2024/Conference — TheWebConf24_

### Official Review · Reviewer_F23H · 2023-11-24

**Novelty:** 5
**Technical Quality:** 4

**Review:**

The paper proposes a novel framework MultiGPrompt for multi-task pre-training and prompting for few-shot learning on graphs.

Strength

1.	The paper proposes a multi-task pre-training and prompting framework for few-shot learning on graphs for the first time.

2.	The Paper is well presented and structured.

3.	The figure of overall framework is clear.


Weakness
1. It seems that the framework design is primarily focused on the node-level and graph-level downstream tasks, i.e., node classification and graph classification. It is unclear whether the effectiveness of MultiGPrompt can be generalized to edge-level tasks (e.g., link prediction).

2. In the experiments, the number of shots varies from 1 to 10. It is suggested to consider larger values, such as commonly used 50 or 100, to comprehensively evaluate the model performance.

**Questions:**

See the weakness.

**Reviewer Confidence:**

3: The reviewer is confident but not certain that the evaluation is correct

**Scope:**

4: The work is relevant to the Web and to the track, and is of broad interest to the community

---

### Official Review · Reviewer_GcCK · 2023-11-27

**Novelty:** 5
**Technical Quality:** 5

**Review:**

The paper proposes MultiGPrompt for multi-task pre-training and prompting in GNNs. To alleviate the reliance on task-specific labeled data, the authors propose pre-training on self-supervised tasks and exploit multiple pretext tasks for more comprehensive pre-trained knowledge. In the pre-training phase, MultiGPrompt uses pretext tokens to synergize diverse tasks, addressing challenges related to task interference. In the downstream adaptation phase, a dual-prompt mechanism, consisting of composed and open prompts, is proposed to transfer both task-specific and global pre-trained knowledge to downstream tasks. Experiments on six benchmark datasets demonstrate the superior performance of MultiGPrompt in few-shot node classification and graph classification compared to existing approaches.


**Strengths:**

(+) The idea of extending the "pre-training and prompting" paradigm to graph learning is an important and popular topic. The proposed multi-task pre-training and prompting framework is also well-motivated.

(+) The presented techniques, including composed and open prompts as well as pretext tokens, are reasonable and intuitive.

(+) The evaluation protocol is comprehensive. Various baselines, including GNNs, graph pre-training models, and graph prompt-based learning models, are compared. The authors also conduct ablation studies and hyperparameter analyses to validate the contribution of their presented techniques and model configuration.

**Weaknesses:**

(-) Experiments are conducted on node classification and graph classification only. It is unclear whether/how the proposed framework can be extended to the link prediction task.

(-) Statistical significance tests are missing. It is not clear whether the gaps between MultiGPrompt and baselines/ablation versions are statistically significant or not. In fact, in Table 2, Table 3, and Figure 5, some gaps are quite subtle (given the large variances), therefore p-values should be reported.

**Questions:**

- Could you show the generalizability of MultiGPrompt by showing its performance on the link prediction task (e.g., on Cora and Citeseer)?

- Could you conduct statistical significance tests to compare MultiGPrompt with the strongest baseline in Table 2 and each ablation version in Table 3/Figure 5?

- Could you provide a theoretical analysis of the number of tunable parameters in MultiGPrompt? This can echo the empirical efficiency analysis in Section 5.4.

**Reviewer Confidence:**

3: The reviewer is confident but not certain that the evaluation is correct

**Scope:**

3: The work is somewhat relevant to the Web and to the track, and is of narrow interest to a sub-community

---

### Official Review · Reviewer_EiVH · 2023-11-27

**Novelty:** 5
**Technical Quality:** 5

**Review:**

The paper proposes a framework to improve the generalization power of GNN by multi-task pretraining and prompting. The pre-training helps model cover more knowledge and the prompting helps model better adapt during inference.

Pros:
The method is interesting and doable in practical industry environments.
The experiment is thorough, showing the significant gain the idea brings.

Cons:
The multi-task pre-training part is limited in novelty.

**Questions:**

Q1: Is there a principled way of choosing tasks and data during the pre-training phase?
Q2: How would the taxonomy in different graphs affect the final quality of the framework?

**Reviewer Confidence:**

1: The reviewer's evaluation is an educated guess

**Scope:**

3: The work is somewhat relevant to the Web and to the track, and is of narrow interest to a sub-community

---

### Decision · Program_Chairs · 2024-01-22

**Decision:**

Accept

**Comment:**

The paper proposes MultiGPrompt for multi-task pre-training and prompting on graphs. The authors leverage several self-supervised tasks for the pre-training, and the experiments demonstrate the effectiveness of the proposed method. During the rebuttal phase, the authors successfully addressed the concerns raised by reviewers, especially for the link prediction task. The feedback from all reviewers has been predominantly positive, reflecting the paper's contribution to the field. Based on the comprehensive review process and the responses provided by the authors, I recommend the acceptance of this paper.